# Relationship between Subjective Oral Discomfort and Health-Related Quality of Life in the South Korean Elderly Population

**DOI:** 10.3390/ijerph17061906

**Published:** 2020-03-15

**Authors:** Kyung-Yi Do, Sook Moon

**Affiliations:** 1Department of Dental Hygiene, Hanseo University, 46 Hanseo 1-ro, Haemi-Myun Seosan-Si, Chungcheognam-do 31962, Korea; dkl8684@naver.com; 2Department of Dental Hygiene, Daejeon Health Institute of Technology, 21, Chungjeong-ro, Dong-gu, Daejeon 34504, Korea

**Keywords:** oral discomfort, toothache, masticatory discomfort, pronunciation problem, health-related quality of life (HRQOL), Korean elderly, Korea National Health and Nutrition Examination Survey (KHNANES)

## Abstract

This study examined the relationship between oral discomfort and health-related quality of life (HRQOL) in the Korean elderly, using the datasets provided by the Korea National Health and Nutrition Examination Survey (KHNANES) over 6 consecutive years (2010–2015). A total of 13,618 participants aged 65 years and over were included in the final analysis. A complex sample logistic regression was performed to determine the impact of oral discomfort on HRQOL. The results revealed that toothache, masticatory discomfort, and pronunciation problems caused by oral health conditions were all risk factors for decreased HRQOL. In particular, masticatory discomfort (adjusted odds ratio (AOR) 1.63, Model III (adjusted for all covariates)) and pronunciation problems (AOR 1.64, Model III) negatively impacted the HRQOL of the elderly to a great extent. Masticatory discomfort had a stronger negative impact on HRQOL in the domains of “self-care” (AOR 1.83) and “usual activities” (AOR 1.66), while pronunciation problems had a similar impact on all five domains of the EuroQol 5-Dimension (EQ-5D). These findings could serve as baseline data for setting up early intervention programs for the timely prevention of oral health-related discomfort problems that greatly affect the QOL of the elderly population, and for the development of comprehensive and efficient dental insurance policies.

## 1. Introduction

Oral health is essential in preserving a healthy life in later years. Thus, oral health in the elderly population plays an important role in maintaining a balanced diet and healthy digestion and is closely related to systemic diseases [1,2]. Furthermore, with age the salivary glands undergo morphological changes, which gradually reduce saliva secretion. Not only does reduced saliva secretion lead to increased vulnerability to dental caries and periodontal diseases in the elderly, but it can also result in oral ulcer and erythema, causing severe discomfort when wearing dentures [3,4]. In addition, oral health has a great impact on self-esteem and quality of life (QOL) in the elderly because it affects pronunciation and physical appearance, which are both of vital importance to social life and interpersonal relationships [5,6]. Finally, tooth loss as a result of oral diseases weakens masticatory function and causes oral discomfort while chewing. In turn, the negative impact on a balanced diet and effective dietary intake ultimately contributes to elevated morbidity and mortality among the elderly population [7,8]. Tooth loss also has a direct impact on health-related quality of life (HRQOL) by causing toothache, inaccurate pronunciation, and altered appearance [7,9].

According to previous studies, toothache caused by oral disease is a potential risk factor for lower QOL, making it difficult for older adults to carry out daily living activities and negatively impacting their mental health, which may result in conditions such as insomnia and depression [9]. On the contrary, a good oral health status helps the elderly talk with confidence to others, restores their self-esteem, and positively impacts their social lives, thus improving their QOL [10,11]. Preventing oral discomfort by promoting oral health is therefore crucial for older adults to maintain a healthy daily life and pursue a high QOL. In this regard, toothache, masticatory discomfort, and pronunciation problems are major risk factors for oral discomforts that deteriorate QOL in the elderly [5,8,9,12]. Timely interventions to treat oral health problems caused by oral diseases may thus play a critical role in relieving these oral discomforts.

However, a majority of previous studies on oral discomfort among older adults have focused exclusively on masticatory dysfunction. Many studies also had limitations, with a small sample size and narrow regional focus, which implied a low degree of generalizability of the research outcomes [11]. Moreover, there are only a limited number of studies examining gender differences in self-rated oral discomfort and QOL. In an effort to address these limitations, this study aimed to investigate the impact of three typical self-rated oral discomfort problems, namely toothache, masticatory discomfort, and pronunciation problems, on the deterioration of HRQOL with a large sample size by combining the data from the Korea National Health and Nutrition Examination Survey (KHNANES) over 6 years (2010–2015).

## 2. Materials and Methods

### 2.1. Participants

This study was conducted with the datasets provided by KHNANES over 6 consecutive years (2010–2015), with the first 3 years (2010–2012) covered in KHNANES V and the next 3 years (2013–2015) in KHNANES VI. KHNANES has been conducted every 3 years from 1998 to 2005 and every year since 2007 in order to provide the basic data needed for the establishment and evaluation of government health policies, such the annual Health Plan and other health-promoting programs. KHNANES has since produced representative and reliable statistics pertaining to the health status, health behaviors, food consumption, and dietary intake of the Korean people at the national and metropolitan/provincial levels.

In a stratified two-stage cluster sampling process, enumeration areas and households were sampled as the primary sample units (PSUs) and secondary sample units (SSUs) in the first and second stages, respectively.

In the fifth KNHANES period (KNHANES V, 2010–2012), 192 enumeration areas were clustered as PSUs every year and a survey was conducted on all members (≥1 year of age) of 3800 households from January to December. In the sixth KNHANES period (KNHANES VI, 2013–2015), 576 enumeration areas were clustered as PSUs every year and 20 households were selected as SSUs from among the households, excluding institutional facilities (e.g., nursing homes, military camps, and prisons) and foreigner households, using systematic sampling.

A total of 48,481 survey participants were included in KNHANES V and VI with the participation of 25,533 out of 31,596 potential survey participants (77.0% participation rate) and 22,948 out of 29,321 potential survey participants (78.3% participation rate) in KNHANES V and VI, respectively. A total of 13,618 participants aged 65 years and over were included in the final analysis of this study. This study was approved by the Institutional Review Board (IRB) of the Daejeon Health Institute of Technology (approval number: 1041490-20190705-HR-004). Further details are available from the Guidelines for the use of KNHANES V (2010–2012) and VI (2013–2015) [13,14].

### 2.2. Measurements

#### 2.2.1. General Demographic Characteristics

General characteristics of the participants included gender, age, educational level, household income level, and diabetes, hypertension, stress, smoking, and drinking status. Age was categorized into 65–74 years, 75–84 years, and 85 years or over; educational level into elementary school, lower, middle, and high school, and college (including community college) or higher; and household income level into low, middle, and high.

Following the classification criteria proposed by the Korean Diabetes Association (KDA), diabetes status was categorized into normoglycemia (fasting glucose <100 mg/dL; blood sugar level for non-diabetics), impaired fasting glucose (100–125 mg/dL; blood sugar level for prediabetes), and diabetes (≥126 mg/dL; diagnosis of diabetes mellitus or current use of hypoglycemic agents or insulin shots) [15,16].

Hypertension was categorized into normal, pre-hypertension, and hypertension in accordance with the hypertension stages in adults aged 19 years or older in Korea as defined in “The Seventh Report of the Joint National Committee on Prevention, Detection, Evaluation, and Treatment of High Blood Pressure” [17]. Normal status was defined as not being prehypertensive or hypertensive and having a systolic blood pressure (SBP) lower than 120 mmHg and/or a diastolic blood pressure (DBP) lower than 80 mmHg. Pre-hypertension was defined as not being hypertensive and having an SBP of 120–139 mmHg and/or a DBP of 80–89 mmHg. Hypertension was defined as an SBP of ≥140 mmHg and/or a DBP of ≥90 mmHg or a history of hypertension medication.

Smoking status was determined by the answer to the question, “Do you currently smoke?” The answers, “I smoke every day” and “I smoke occasionally” were reclassified as “yes”, and “I have quit smoking” was reclassified as “no.” Drinking status was determined using the yes/no question, “Have you ever had an alcoholic drink?”

#### 2.2.2. Oral Discomfort

Oral discomfort, the independent variable of this study, was broken down into three sub-variables: a history of toothache, masticatory discomfort, and pronunciation problems within the previous year. A history of toothache was determined by the answer to the yes/no question, “Have you ever had a toothache (throbbing pain in or around a tooth or painful sensation in the teeth when taking hot or cold foods and beverages)?” Masticatory discomfort was determined by the answer to the question, “Do you feel uncomfortable when chewing food due to problems in your mouth (teeth, denture, or gum)?” The answers “Very uncomfortable” and “uncomfortable” were reclassified as “yes,” and “more or less,” “not uncomfortable,” and “not uncomfortable at all” were reclassified as “no.” Pronunciation problems were determined by the answer to the question, “Do you have difficulty or discomfort in your pronunciation due to problems in your mouth (teeth, denture, or gum)?” The answers “Very uncomfortable” and “uncomfortable” were reclassified as “discomfort,” “more or less” as “moderate,” and “not uncomfortable” and “not uncomfortable at all” as “comfort.”

#### 2.2.3. Health-Related Quality of Life (EQ-5 Dimension)

For the assessment of HRQOL, we used the Korean version of the EuroQol 5-Dimension (EQ-5D). The EQ-5D comprises five dimensions: (1) usual activities, (2) self-care, (3) anxiety/depression, (4) pain/discomfort, and (5) physical activities. The internal consistency reliability of the entire scale was good (Cronbach’s α value: 0.793). In this study, each of the five dimensions was rated on a three-point scale (1 = no problem, 2 = moderate problem, and 3 = extreme problem), with the total score ranging from 1 to 15 points, whereby a lower total score indicated a higher QOL and vice-versa. The total score was evaluated as either “good” (1–5 points) or “not good” (6–15 points). For subscale analysis of each of the five dimensions, “no problem” was rated as “good” and “moderate problem” and “extreme problem” were rated as “not good.”

### 2.3. Statistical Analysis

The KNHANES data were collected according to a complex sample design, adopting the stratified two-stage cluster sampling approach in order to draw nationally representative data. Therefore, we conducted all statistical analyses according to a complex sample design procedure by applying stratification variable, cluster variable (PSU), and sample weights. Rao-Scott chi-square test was performed to examine the association between participants’ general characteristics, oral discomfort, and HRQOL. Next, a complex sample logistic regression was performed to examine the impact of oral discomfort on HRQOL. Model I presented crude odds ratio (OR) and 95% confidence interval (CI). Model II was applied to estimate the adjusted odds ratio (AOR) and 95% CI after adjusting for gender and age. Model III presented AOR and 95% CI after adjusting for all covariates to examine the true impact of oral discomfort on HRQOL. In order to examine the impact of oral discomfort on each dimension of the Euro-5D, a logistic regression analysis was performed after adjusting for all covariates, and AOR and 95% CI were presented. Finally, to examine gender differences in the impact of oral discomfort on HRQOL, a complex sample logistic regression was performed separately on male and female subgroups. All analyses were performed using PASW Statistics 18.0. (IBM Corp., Armonk, NY, USA), and statistical significance was set at *p* < 0.05.

## 3. Results

### 3.1. Health-Related Quality of Life (EQ-5D Score) according to General Characteristics and Oral Discomfort Problems

When analyzing the participants’ general characteristics, the number of elderly respondents who perceived their HRQOL as being “not good” was higher among women than men. Moreover, HRQOL decreased as age increased (*p* < 0.001), educational level decreased (*p* < 0.001), and income level decreased (*p* < 0.001). A lower HRQOL was associated with the presence of diabetes (*p* < 0.005) and hypertension (*p* < 0.001), a higher stress level (*p* < 0.001), and current smoker and alcohol user status (*p* < 0.001). In terms of oral discomfort, toothache, masticatory discomfort, and pronunciation problems were found to be risk factors for a lower HRQOL (*p* < 0.001 for all) (Table 1).

### 3.2. Association between Oral Discomfort Problems and Health-Related Quality of Life (EQ-5D Score)

A complex sample logistic regression was performed to examine the impact of toothache, masticatory discomfort, and pronunciation problems on HRQOL. In Model I, estimating the crude OR before adjustment for covariates, higher crude ORs were found in participants with a history of toothache (odd ratio (OR) = 1.18, 95% confidence interval (CI) = 1.04–1.34), masticatory discomfort (OR = 2.26, 95% CI = 2.02–2.53), and pronunciation problems (OR = 1.81, 95% CI = 1.53–2.14) compared with participants without oral discomfort problems. In Model II for estimating the AOR adjusted for gender and age, higher AORs were found in participants with a history of toothache (AOR = 1.25, 95% CI = 1.09–1.42), masticatory discomfort (AOR = 1.71, 95% CI = 1.50–1.96), and pronunciation problems (AOR = 1.86, 95% CI = 1.56–2.20) compared with participants without these oral discomfort problems, thus verifying them as risk factors for low HRQOL. In Model III for estimating the AOR adjusted for all covariates, higher AORs were found in participants with a history of toothache (AOR = 1.20, 95% CI = 1.04–1.39), masticatory discomfort (AOR = 1.63, 95% CI = 1.40–1.89), and pronunciation problems (AOR = 1.64, 95% CI = 1.36–1.97) compared with participants without oral discomfort problems. Although the intergroup differences in Model III were observed to be lower than in Model I and Model II, these risk factors were still found to have a negative impact on HRQO. In particular, masticatory discomfort and pronunciation problems exerted a greater impact on the HRQOL of the elderly than toothache (Table 2).

### 3.3. Association between Oral Discomfort Problems and Five EQ-5D Dimensions

To assess the impact of oral discomfort on HRQOL in each of the five dimensions of EQ-5D, a complex sample logistic regression analysis was performed after adjusting for all covariates. As a result, higher AORs were found in participants with a history of toothache in “usual activities” (AOR = 1.19, 95% CI = 1.01–1.39), “anxiety/depression” (AOR = 1.36, 95% CI = 1.15–1.61), and “physical activities” (AOR = 1.17, 95% CI = 1.02–1.34) compared with participants without it, indicating that toothache has the greatest negative impact on “anxiety/depression” among the five dimensions of EQ-5D. Slightly higher AORs were observed in “self-care” and “pain/discomfort,” albeit without statistical significance. Moreover, higher AORs were found in patients with masticatory discomfort compared with patients without it in all five dimensions of EQ-5D: “usual activities” (AOR = 1.66, 95% CI = 1.39–1.98), “self-care” (AOR = 1.83, 95% CI = 1.46–2.30), “anxiety/depression”(AOR = 1.43, 95% CI = 1.19–1.71), “pain/discomfort” (AOR = 1.58, 95% CI = 1.36–1.84), and “physical activities” (AOR = 1.53, 95% CI = 1.32–1.78), indicating that masticatory discomfort has the greatest negative impact on the EQ-5D dimension “self-care.” Similarly, for pronunciation problems, higher AORs were found in participants who answered “discomfort” compared with participants who answered with “comfort” and “moderate” in all five dimensions of EQ-5D: “usual activities” (AOR = 1.60, 95% CI = 1.32–1.94), “self-care” (AOR = 1.55, 95% CI = 1.22–1.97), “anxiety/depression” (AOR = 1.55, 95% CI = 1.23–1.94), “pain/discomfort” (AOR = 1.44, 95% CI = 1.20–1.72), and “physical activities” (AOR = 1.52, 95% CI = 1.26–1.83). These results confirmed that all three oral health discomfort problems had a negative impact on all five domains of HRQOL. In particular, masticatory discomfort and pronunciation problems were identified as risk factors for HRQOL in the EQ-5D domains of “usual activities” and “self-care” (Table 3).

### 3.4. Subgroup Analysis for Association between Oral Discomfort Problems and Health-Related Quality of Life (EQ-5D Score) by Gender

A subgroup analysis was conducted on male and female subgroups to investigate gender differences in the negative impact of oral discomfort on HRQOL.

In the male subgroup, the crude ORs were slightly higher in participants with a history of toothache than in participants without such a history (OR = 1.19, 95% CI = 1.01–1.41), but the intergroup differences did not reach statistical significance in Models II and III. In Model III estimating the AOR adjusted for all covariates, higher AORs for lower HRQOL were found in male participants with a history of masticatory discomfort (AOR = 1.53, 95% CI = 1.32–1.78) and pronunciation problems (AOR = 1.44, 95% CI = 1.11–1.87) compared with male participants without oral discomfort problems.

In the female subgroup, the AORs for lower HRQOL (Model III) were higher in participants with a history of toothache (OR = 1.29, 95% CI = 1.06–1.57) and pronunciation problems (AOR = 1.84, 95% CI = 1.43–2.38) than in female participants without oral discomfort problems. This finding suggested that a history of toothache and pronunciation problems in the elderly exerted a greater impact on HRQOL in women than in men. Higher AORs for lower HRQOL were similar to those in the male subgroup (AOR = 1.52, 95% CI = 1.16–1.99).

In the subgroup analysis, masticatory discomfort and pronunciation problems were found to have the greatest negative impact on HRQOL in the male and female subgroups, respectively, demonstrating gender differences in the components of oral discomfort that have a negative impact on HRQOL among the elderly (Table 4).

## 4. Discussion

With the global aging of the population and increasing life expectancies, the issue of QOL in the elderly population, i.e., how people can remain healthy in later years, has gained great attention [6,11,18].

The results of this study revealed that toothache, masticatory discomfort, and pronunciation problems caused by oral diseases were risk factors for decreased HRQOL. All three oral discomforts, particularly masticatory discomfort and pronunciation problems had an adverse impact on the HRQOL of the elderly. The negative impact of masticatory discomfort on HRQOL was stronger in the domains of “self-care” and “usual activities”. In contrast, pronunciation problems had a similar impact on all five domains of the EQ-5D, even though it tended to be slightly higher in the domain of “usual activities”.

Older people are especially prone to chronic oral diseases, such as dental caries and periodontal diseases. Without appropriate treatment, such oral conditions may trigger not only systemic diseases but also a variety of other problems including mental health problems and depression, aesthetic issues, and interpersonal relationships, thus deteriorating the QOL of the elderly [5,7,12,19,20]. Moreover, oral diseases restrict dietary intake, which is the most crucial factor in maintaining good health in older people, leading to undernutrition and, ultimately, the triggering of systemic diseases and exacerbation of other health conditions [7,21,22,23]. Therefore, improving various oral discomfort problems through the timely treatment of oral diseases is crucial in boosting the QOL of the elderly population.

A previous study reported that elderly people with a lower chewing ability are 2.36 times more likely to have poor HRQOL compared to those without [12]. Another study reported that toothache in older people restricts their ability to carry out daily living activities, and is associated with a 1.5-fold higher risk of mental disorders, such as anxiety, depression, and suicidal ideation, compared with older people without toothache [9]. Likewise, our analysis of the association between oral discomfort problems and the five dimensions of the EQ-5D revealed that older people with toothache have 1.3 times higher odds for anxiety/depression versus those without, which is consistent with previous research. Furthermore, masticatory discomfort was found to deteriorate the QOL in all five domains of EQ-5D, and more particularly in the domains of “self-care” and “pronunciation problems” with 1.83- and 1.5-times higher odds, respectively. Furthermore, pronunciation problems were found to negatively impact the QOL of the elderly more severely in women than men.

Oral discomfort caused by oral diseases can be adequately addressed by appropriate dentures and prosthetic treatment, including implants [24]. However, it has also been reported that a great proportion of the elderly population are excluded from receiving prosthetic treatment due to their vulnerable economic conditions [25,26,27]. Other studies have reported that the intensity and duration of experiencing oral discomfort may differ depending on gender, socioeconomic level, and individual oral environment [5,28]. Altogether, more than 50% of the worldwide elderly population suffers from masticatory discomfort, with more women affected than men [27]. According to a Korean study, more than 30% of the elderly population in Korea uses dentures, with the percentage being slightly higher among women than men, and women are more sensitive to the use of dentures and feel discomfort more intensely than men [5]. Our subgroup analysis examining gender differences revealed that toothache and pronunciation problems had a stronger negative impact on the QOL of the elderly for women than men. Masticatory discomfort and pronunciation problems were found to have the greatest impact on the QOL of elderly men and women, respectively, thus confirming gender differences in perceiving these oral discomfort problems.

Socioeconomically disadvantaged people often tend to fail to receive timely treatment and leave their oral diseases unattended for a prolonged period of time, which aggravates their oral conditions, causing them to live with oral discomfort problems throughout their lifetime, which, in turn, seriously deteriorates their QOL in later years [25,26,27]. It has been reported that socioeconomic inequality among older people still exists, albeit to a lesser extent, and that older people who are economically disadvantaged or living in rural areas with limited access to healthcare services experience oral discomfort, such as masticatory discomfort, more frequently than those who are not [5,29,30]. Consistently, in our study, respondents with lower income levels were found to have a lower QOL than those with higher income levels by two-fold. Older people’s access to dental treatment services may vary according to their socioeconomic levels, family status, and self-rated health status [27]. It has also been reported that economically inactive elderly individuals experience masticatory discomfort more frequently [5,27]. Efforts will thus have to be made to pursue efficient and comprehensive dental insurance policies extending to the socioeconomically disadvantaged elderly with a view to promoting the early prevention of oral diseases or relief of oral discomfort, thus improving their QOL.

Due to its cross-sectional design, this study is limited in determining the direct causal relationships between the independent and dependent variables. Other limitations are the potential information bias associated with using a self-rated questionnaire in examining oral discomfort problems, which are the independent variables of this study, and the failure to determine the intensity of toothache and masticatory discomfort, because these were examined based on yes/no questions

We were also unable to adjust compounding factors such as oral disease (periodontitis, dry mouth, etc.) that may affect oral discomfort. Therefore, we could not control for all confounding variables, which may have influenced the magnitude of the study results. Therefore, more elaborate research is necessary in follow-up studies, utilizing an instrument with higher internal consistency and reliability in order to better quantify the oral discomfort problems that can have an impact on the QOL of the elderly population. Despite these limitations, one of this study’s strengths is the generalizability of the findings: it utilized a nationally representative dataset encompassing 6 years’ worth of data spanning the KNHANES V and VI periods, such that its findings can be generalized to the Korean elderly population aged 65 years and over. Furthermore, we established the reliability of this study by using the EQ-5D, an HRQOD instrument with proven reliability and validity, for the assessment of QOL, which is the dependent variable of this study. This study differentiates itself from previous studies in that it examined the association between QOL and three typical oral health indicators (toothache, masticatory discomfort, and pronunciation problems) rather than a single oral discomfort problem. We also verified gender differences in the negative impact of the individual oral discomfort problems on the QOL of the elderly and their gender-specific intensities by performing a subgroup analysis. The findings of this study are expected to serve as baseline data for setting up early intervention programs for the timely prevention of oral-health-related discomfort problems that greatly affect the QOL of the elderly population and the development of comprehensive and efficient dental insurance policies.

## 5. Conclusions

We confirmed that toothache, masticatory discomfort, and pronunciation problems caused by oral diseases were risk factors for decreased HRQOL, as well as confirming gender differences in perceiving these oral discomfort problems. These findings could serve as baseline data for setting up early intervention programs for the timely prevention of oral health-related discomfort. Subsequent studies need a well-designed longitudinal study that can clarify the causal relationship between oral discomforts and HRQOL.

## Figures and Tables

**Table 1 ijerph-17-01906-t001:** Health-related quality of life (EQ-5D score) according to general characteristics and oral discomfort problems (*n* = 13,618).

Variable	Category	EQ-5D *n* (Weighted %)		SE	*p*-Value
		Good	Not Good		
Gender	Male	3231 (54.4)	2628 (45.6)	1.1	<0.001 ***
	Female	2646 (32.8)	5113 (67.2)	0.8	
Age	65–74	4303 (47.1)	4736 (52.9)	0.9	<0.001 ***
	75–84	1504 (32.9)	2853 (67.1)	1.2	
	≥85	70 (31.7)	152 (68.3)	5.6	
Education level	≥University	692 (70.2)	292 (29.8)	2.4	<0.001 ***
	High school	1170 (58.4)	854 (41.6)	1.7	
	Middle school	853 (48.2)	858 (51.8)	1.9	
	≤Elementary school	3149 (34.8)	5702 (65.2)	0.8	
Household Income level	High	738 (56.4)	599 (43.6)	2.1	<0.001 ***
	Middle	2636 (47.8)	2685 (52.2)	1.1	
	Low	2440 (34.6)	4378 (65.4)	1	
Diabetes	Normoglycemia	2653 (43.5)	3332 (56.5)	1.1	0.005 **
	Impaired fasting glucose	1401 (47.0)	1507 (53.0)	1.5	
	Diabetes	1081 (40.1)	1522 (59.9)	1.5	
Hypertension	Normal	1082 (44.9)	1236 (55.1)	1.7	<0.001 ***
	Pre-Hypertension	1297 (45.5)	1464 (54.5)	1.5	
	Hypertension	3486 (39.7)	5028 (60.3)	0.9	
Stress level	High	680 (23.6)	2087 (76.4)	1.2	<0.001 ***
	Low	5138 (47.0)	5529 (53.0)	0.8	
Alcohol	Yes	4395 (44.7)	5188 (55.3)	0.9	<0.001 ***
	No	1438 (35.4)	2456 (64.6)	1.2	
Smoking	Yes	2796 (50.4)	2629 (49.6)	1.1	<0.001 ***
	No	3025 (36.5)	4995 (63.5)	0.9	
**Oral discomfort**					
Toothache	Yes	1566 (36.6)	2584 (63.4)	1.2	< 0.001 ***
	No	4097 (44.3)	4859 (55.7)	0.9	
Masticatory discomfort	Yes	1961 (31.4)	4052 (68.6)	0.9	<0.001 ***
	No	3829 (50.9)	3550 (49.1)	1	
Pronunciation problems	Comfort	4143 (48.1)	4213 (51.9)	0.9	<0.001 ***
	Moderate	813 (39.4)	1224 (60.6)	1.8	
	Discomfort	821 (27.2)	2160 (72.8)	1.3	

The data were analyzed by Rao-scott chi-square test for complex sample. Significance level, *** *p* < 0.001, ** *p* < 0.01.

**Table 2 ijerph-17-01906-t002:** Association between oral discomfort problems and health-related quality of life (EQ-5D score).

Variable	Model I ^a^ OR (95% CI)	Model II ^b^ AOR (95% CI) ^d^	Model III ^c^ AOR (95% CI)
Toothache			
No	1	1	1
Yes	1.18 (1.04–1.34)	1.25 (1.09–1.42)	1.20 (1.04–1.39)
Masticatory discomfort			
No	1	1	1
Yes	2.26 (2.02–2.53)	1.71 (1.50–1.96)	1.63 (1.40–1.89)
Pronunciation Problems			
Comfort	1	1	1
Moderate	1.25 (1.05–1.49)	1.28 (1.08–1.51)	1.26 (1.04–1.52)
Discomfort	1.81 (1.53–2.14)	1.86(1.56–2.20)	1.64 (1.36–1.97)

The data were analyzed by logistic regression for complex sample. ^a^ Model I: Unadjusted OR (95% CI). ^b^ Model II: Adjusted for gender, age. ^c^ Model III: Adjusted for all covariates (gender, age, education level, household income level, diabetes, hypertension, stress level, alcohol, smoking). ^d^ AOR (95% CI): Adjusted Odds Ratio (95% confidence Interval).

**Table 3 ijerph-17-01906-t003:** Logistic regression analysis for association between oral discomfort problems and five dimensions of EQ-5D.

Variable	Category	Usual Activities	Self-Care	Anxiety/Depression	Pain/Discomfort	Physical Activities
		AOR (95% CI)	AOR (95% CI)	AOR (95% CI)	AOR (95% CI)	AOR (95% CI)
Toothache	No	1	1	1	1	1
	Yes	1.19 (1.01–1.39)	1.15 (0.94–1.41)	1.36 (1.15–1.61)	1.15 (1.00–1.33)	1.17 (1.02–1.34)
Masticatory discomfort	No	1	1	1	1	1
	Yes	1.66 (1.39–1.98)	1.83 (1.46–2.30)	1.43 (1.19–1.71)	1.58 (1.36–1.84)	1.53 (1.32–1.78)
Pronunciation problems	Comfort	1	1	1	1	1
	Moderate	1.25 (0.99–1.58)	1.30 (0.98–1.71)	1.25 (0.98–1.59)	1.25 (1.03–1.51)	1.34 (1.10–1.63)
	Discomfort	1.60 (1.32–1.94)	1.55 (1.22–1.97)	1.55 (1.23–1.94)	1.44 (1.20–1.72)	1.52 (1.26–1.83)

The data were analyzed by logistic regression for complex sample. Dependent variables (EQ-5D, EuroQol 5-Dimension) reference is all “Good”. AOR—adjusted odds ratio, Adjusted for all covariates (gender, age, education level, household income level, diabetes, hypertension, stress level, alcohol, smoking); 95% CI—95% confidence interval.

**Table 4 ijerph-17-01906-t004:** Subgroup analysis for association between oral discomfort problems and health-related quality of life (EQ-5D score) by gender.

Variable	Model I ^a^ OR (95% CI)	Model II ^b^ AOR (95% CI) ^d^	Model III ^c^ AOR (95% CI)
*Men*			
Toothache			
No	1	1	1
Yes	1.19 (1.01–1.41)	1.10 (0.91–1.31)	1.10 (0.91–1.34)
Masticatory discomfort			
No	1	1	1
Yes	2.14 (1.81–2.54)	1.62 (1.25–2.09)	1.53 (1.16–2.02)
Pronunciation problems			
Comfort	1	1	1
Moderate	1.31 (1.04–1.65)	1.15 (0.90–1.48)	1.24 (0.93–1.66)
Discomfort	2.23 (1.80–2.76)	1.59 (1.25–2.03)	1.44 (1.11–1.87)
*Women*			
Toothache			
No	1	1	1
Yes	1.56 (1.32–1.85)	1.37 (1.14–1.63)	1.29 (1.06–1.57)
Masticatory discomfort			
No	1	1	1
Yes	2.29 (1.94–2.69)	1.40 (1.10–1.78)	1.52 (1.16–1.99)
Pronunciation problems			
Comfort	1	1	1
Moderate	1.45 (1.18–1.80)	1.23 (0.98–1.56)	1.21 (0.93–1.58)
Discomfort	2.86 (2.32–3.50)	2.06 (1.64–2.59)	1.84 (1.43–2.38)

The data were analyzed by logistic regression for complex sample. ^a^ Model I: Unadjusted OR (95% CI). ^b^ Model II: Adjusted for age. ^c^ Model III: Adjusted for all covariates (age, education level, household income level, diabetes, hypertension stress level, alcohol, smoking). ^d^ AOR (95% CI): Adjusted Odds Ratio (95% confidence Interval).

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
