# Peer review of "Relationship between Subjective Oral Discomfort and Health-Related Quality of Life in the South Korean Elderly Population"

_ijerph, 2020, doi:10.3390/ijerph17061906_

Round 1

Reviewer 1 Report

I believe authors need to avoid using general recommendation as this study was conducted in a single study centre which affects external validity of the study. Authors need to explain this as a limitation. 

Author Response

Response to Reviewer 1 Comments

Point 1. I believe authors need to avoid using general recommendation as this study was conducted in a single study centre which affects external validity of the study. Authors need to explain this as a limitation. 

Response 1.

Thank you for your comments.

This study is a cross-sectional survey based on representative national data extracted from the population of Korea.

Therefore, the greatest advantage of the study is that it can generalize the results of the research to the whole population, and based on these findings, it can be provided as a basis for developing national health policy.

Basic information on the National Health and Nutrition Survey is described below.

The National Health and Nutrition Survey consists of a screening survey, a health survey, and a nutrition survey.

This is a statutory investigation aimed at producing statistically representative and reliable statistics on national health level, health-related consciousness and behavior, and food and nutrition intake. In particular, statistical data is used as an indicator for setting and evaluating goals of the Korea’s Health Plan, and is used as basic data for establishing and evaluating health policies such as developing health promotion programs. It also produces health indicators provided to international organizations such as the WHO and the OECD, and is introduced in countries such as the USA’s National Health and Nutrition Examination Survey and Japan’s National Health and Nutrition Survey. The index is comparable with the results of the national survey.

For more details, please refer to the National Health and Nutrition Survey Instruction Manual.

I think the fundamental limitations of this study are fully described in the discussion part.

Acknowledgements

Thank you for your careful review of my manuscript.
I also appreciate your advice on helping me improve my paper.
We will present a better paper to your journal in the future.

Reviewer 2 Report

You should need a general revision of English and a revision within the cell in the tables that conforms according to the rules.

In your paper, you should explain what kind of insurance efficiency is in terms of a dental insurance policy.

The author's research has already been proven through existing literature research.
What content do you want to convey through data this paper?

Could you let me know the difference between small sample size and large sample size to be the purpose of the study?

The number of participants in the study is incorrect.
Please refer to 83 to 89 lines and explain.

According to the results of your data, look at Table 1. If elderly people don't have any experience as smokers and alcohol users in the study, what do you think about the study result that EQ-5D is NOT GOOD?

Why is the quality of life of elderly women lower than that of men by various variables, as shown in Table 4? What do you think about this result?

Please answer the research results.

I hope you have good results.

Author Response

Response to Reviewer 2 Comments

You should need a general revision of English and a revision within the cell in the tables that conforms according to the rules.

Point 1.In your paper, you should explain what kind of insurance efficiency is in terms of a dental insurance policy.

Response 1.

Thank you for the good comments.

The effective dental insurance policy mentioned in this study means that the dental insurance policy, which can be provided at the same age regardless of economic conditions, should be differentially provided in consideration of economic conditions.

The reason is that the economic conditions of the elderly in Korea are wide, and the elderly with poor economic conditions may suffer from oral discomfort for a long time because they do not receive dental treatment in a timely manner.

This is described in the discussion part lines 311-322.

Point 2. The author's research has already been proven through existing literature research.
What content do you want to convey through data this paper?

Response 2.

Similar studies are being conducted continuously to further solidify the scientific basis for the results of previous studies, and studies that complement the limitations of the existing studies gradually strengthen the relationship between the two factors. In this study, an additional investigation was conducted on "pronunciation problems" due to oral problems that were not covered by previous studies, and the fact that the differences between male and female elderly were studied was not suggested in the previous studies. Another difference is that the quality of life has been studied in detail in five dimension . This study conducted a large-scale sample using six years of data to produce a reliable result that oral discomfort in Korean elderly could degrade their quality of life. The strongest advantage is that the results can be used as the basis for the development of national health policy.

Point 3. Could you let me know the difference between small sample size and large sample size to be the purpose of the study?

Response 3.

The difference between studies with large samples is that the results of the study can be trusted and the results can be generalized to the entire population.

Point 4. The number of participants in the study is incorrect.
Please refer to 83 to 89 lines and explain.

Response 4.

A total of 48,481 people is actually the sum of the participants in the survey.

Related contents clearly states in the manuscript as below

A total of 48,481 survey participants were included in KNHANES V and VI with the participation of 25,533 out of 31,596 potential survey participants (77.0% participation rate) and 22,948 out of 29,321 potential survey participants (78.3% participation rate) in KNHANES V and VI, respectively. A total of 13,618 participants aged 65 years and over were included in the final analysis of this study.

Point 5. According to the results of your data, look at Table 1. If elderly people don't have any experience as smokers and alcohol users in the study, what do you think about the study result that EQ-5D is NOT GOOD?

Response 5.

The relationship between smoking and drinking variables and quality of life are not key points in this study. In our opinion, you may feel that moderate smoking and drinking relieve the stress of the subject in the quality of life that you feel subjectively.

It seems that this eventually influenced the quality of life that we felt subjectively.

However, I think this problem should be done in further study in more detail.

Point 6. Why is the quality of life of elderly women lower than that of men by various variables, as shown in Table 4? What do you think about this result?

Response 6.

Studies have shown that women are more sensitive to stress than men and more discomfort like oral pain. These results may explain why women feel lower quality of life than men.

This is described in discussion part lines 298-302.

Acknowledgements

Thank you for your careful review of my manuscript.
I also appreciate your advice on helping me improve my paper.
We will present a better paper to your journal in the future.

Reviewer 3 Report

The manuscript describes the relationship between oral discomfort and health-related quality of life. The topic is very interesting and, although this relation is just well-established and evaluated in previous studies, the manuscript has the strenght of the number of sample, that is very high. 

The limitations of the study are clearly explained especially in the confounding variables that can interfere with the evaluation. 

PLease check the list of the references that not follows a right order (35 references are listed iinsted of 30 references reported in the text). 

Author Response

Response to Reviewer 3 Comments

The manuscript describes the relationship between oral discomfort and health-related quality of life. The topic is very interesting and, although this relation is just well-established and evaluated in previous studies, the manuscript has the strenght of the number of sample, that is very high. 

Point 1.The limitations of the study are clearly explained especially in the confounding variables that can interfere with the evaluation. 

Response 1.

Thanks for the good comments.

We added and revised the related information to reflect the reviewer's comment.

Please check in the discussion part lines 311-322.

Point 2. PLease check the list of the references that not follows a right order (35 references are listed iinsted of 30 references reported in the text). 

Response 2.

We put together 30 references cited in the text from the original manuscripts into an endnote program and submitted the correct version.

However, it seems that an error occurred due to a mistake in the editing stage at the journal.

So we checked again and corrected the reference back to the manuscripts

Acknowledgements

Thank you for your careful review of my manuscript.
I also appreciate your advice on helping me improve my paper.
We will present a better paper to your journal in the future.

Round 2

Reviewer 2 Report

I checked the revised paper well. I also confirmed your response.

In lines 56 to 57, the sample size is considered to be the author's personal opinion, and references are required.

There are overlapping English sentences between 338 and 345, and additional general corrections are required.

You did a really good job. I hope you have a good result.

Thank you